∂ | **Open Peer Review** | Applied and Industrial Microbiology | Research Article

# Joining the bacterial conversation: increasing the cultivation efficiency of soil bacteria with acyl-homoserine lactones and cAMP

**Marco A. Lopez Marin,**[1,2] **Michal Strejcek,**[1] **Ondrej Uhlik**[1]

**ABSTRACT** Bacteria need to be isolated in pure culture to gain access to the wide array of interesting functions they conceal. This is challenging because they do not live in isolation but instead in communities where they actively communicate and interact with each other. In this study, we aimed to increase the culturability of soil bacteria using three different signaling molecules: N-(3-oxohexanoyl)-L-homoserine lactone, N-octanoyl-L-homoserine lactone, and 3',5'-cyclic adenosine monophosphate (cAMP). The signals were added individually to soils suspended in PBS buffer to a final concentration of 5 µM. Soil suspensions were agitated for 24 hours, after which they were serially diluted and plated on tenfold-diluted Reasoner's 2A agar that either contained or did not contain the same signaling molecule used during the extraction step. DNA was isolated from both soil suspensions and grown cultures and, after a high-throughput amplicon sequencing, differences in bacterial abundances and diversity were determined across treatments. To further explain the action of the signaling molecules on the treated soil communities, their metagenomic functions were predicted using the software PICRUSt2. N-octanoyl-L-homoserine lactone was found to increase the diversity observed on solid media, while N-(3-oxohexanoyl)-L-homoserine lactone and cAMP were not. Potentially novel isolates aided by the signaling molecules were affiliated with the genera *Pseudomonas* and *Nocardioides*.

**IMPORTANCE** Microorganisms are a repository of interesting metabolites and functions. Therefore, accessing them is an important exercise for advancing not only basic questions about their physiology but also to advance technological applications. In this sense, increasing the culturability of environmental microorganisms remains an important endeavor for modern microbiology. Because microorganisms do not live in isolation in their environments, molecules can be added to the cultivation strategies to "inform them" that they are present in growth-permissive environmental conditions. Signaling molecules such as acyl-homoserine lactones and 3',5'-cyclic adenosine monophosphate belong to the plethora of molecules used by bacteria to communicate with each other in a phenomenon called quorum sensing. Therefore, including quorum sensing molecules can be an incentive for microorganisms, specifically soil bacteria, to increase their numbers on solid media.

**KEYWORDS** signaling molecules, acyl-homoserine lactones, cAMP, non-culturable bacteria, increased culturability, oligotrophic medium

G rowing bacteria in pure cultures is far from old-fashioned. Technologies may improve, but characterizing the natural world will never become passé because life's dark matter still conceals many secrets worth uncovering. Such concealed potential includes the discovery of novel bioactive molecules or the optimization of

Address correspondence to Marco A. Lopez Marin, marco.antonio.lopez.marin@vscht.cz.

The authors declare no conflict of interest.

See the funding table on p. 12.

microbial-based processes such as bioremediation (1). Therefore, efforts to culture the vast majority of hitherto uncultured prokaryotes are still relevant.

One of the pitfalls of pure cultures is precisely what the concept entails. Just as an animal will probably deteriorate in a zoological garden if kept in isolation, a bacterium may never grow in an artificial laboratory environment isolated from the rest of the organisms of its natural community. Microorganisms are social organisms that form complex communities (2) where they communicate with one another through chemical signals (3). Some of the most important signaling compounds in bacteria are acyl-homoserine lactones (AHLs). The AHL concentration in a specific environment is a proxy of the population size, or cell density (4), which allows bacteria to assess their numbers and thus initiate collective actions such as biofilm formation (5), the production of virulence factors (6, 7), swimming and swarming motility (8), expression of extracellular degradative enzymes (9), or the expression of protective enzymes such as superoxide dismutase and catalase (10).

3',5'-cyclic adenosine monophosphate (cAMP) is a nucleotide second messenger that regulates gene expression and has a role in several important bacterial functions similar to those influenced by AHLs, such as biofilm formation, flagellum synthesis, enterotoxin production, filamentation, and many others (11). In *Pseudomonas aeruginosa*, for example, cAMP joins the virulence factor regulator (Vfr), a regulatory transcription factor responsible for motility, virulence, biofilm formation, and pathogenicity (12). Finally, cAMP can also be a stimulus for dormant cells to resume growth (13).

Culturing bacteria directly in their natural environment where these *messenger molecules* are present increases cultivation efficiency (14). It can therefore be inferred that including signaling compounds in the growth medium can increase the number of culturable cells. This was proved by Bruns et al. (13), who included cAMP, N-butyryl homoserine lactone, and N-oxohexanoyl-DL-homoserine lactone at low concentrations (10 µM) to increase the cultivation efficiency of bacteria from the Baltic Sea. The most effective signaling molecule with the highest cultivation efficiencies was cAMP, but the AHLs added also enhanced the culturability (13). The cAMP also increased the cultivation efficiency of bacterioplankton from Lake Zwischenahner by 10% (15). Finally, some bacterial clades, such as *Sphingobacteria*, can increase their relative abundance when the growth medium is supplemented with AHLs (16). To our knowledge, no data have been published on the effect of cAMP and AHLs on the culturability of soil bacteria.

The purpose of this study was to increase the culturability of soil bacteria on a solid medium using three different signaling molecules independently: cAMP, N-(3 oxohexanoyl)-L-homoserine lactone, and N-octanoyl-L-homoserine lactone. Each signaling molecule was included either during the 24 hours extraction with PBS buffer, on the 10-fold diluted Reasoner's 2A agar plates used for cultivation, or both during the extraction and cultivation. This design allowed, using a high-throughput 16S rRNA approach, to determine how the signaling molecules altered the native soil community, which members of this community benefited from the addition of signaling molecules and when the addition of signaling molecules was effective for increasing culturability.

## MATERIALS AND METHODS

### Extraction of bacteria from soil using cAMP and two different AHLs

The soil used in this study was collected from a garden compost in Mirošovice, Central Bohemia, Czech Republic. It was a sandy loam with characteristics that have been described previously (17). The soil was in refrigeration for 4 years to induce dormancy, and sieved through a 1 mm filter before use.

N-(3-oxohexanoyl)-L-homoserine lactone (AHL1), N-octanoyl-L-homoserine lactone (AHL2), and 3',5'-cyclic adenosine monophosphate (cAMP) were added during the extraction of bacteria from soil. A gram of filtered soil was mixed with 9 mL of PBS buffer (pH 7.4) containing 5 µM of either of the signaling molecules. Both AHLs were first diluted in ethyl ether before they were mixed with the PBS-soil suspension. As a control

for the use of ethyl ether in the extraction procedure, an extraction with 9 mL of buffer and 0.28% ethyl acetate (vol/vol) was included. These samples with solvent are further referred to as "Control_solvent". The water-soluble cAMP was added directly to the PBS-soil suspension to reach a concentration of 5 µM. An extraction with only PBS-soil suspension ("control_no_solvent") was included as a control for the cAMP extraction. Each extraction was carried out in triplicates. The suspensions were agitated for 1 day at 120 rpm and 28°C. A non-agitated PBS-soil suspension was also used as a control (in triplicates). In this case, DNA was isolated immediately after the soil was mixed with PBS (see further, DNA isolation). This control is referred further to as "original soil". In total, 18 soil suspensions were prepared from which DNA was extracted: three for each of the three signaling molecules, three suspensions for two controls (with and without solvent), and three original soil suspensions.

## Solid medium preparation

Tenfold-diluted Reasoner's 2A agar (R2A, Himedia Laboratories, India) was used in this study for culturing the agitated PBS-soil suspensions. Each of the three signaling molecules was added to the R2A individually (5 µM) just before it solidified. AHLs were added to the medium diluted in ethyl ether. R2A plates without signaling molecules (control plates) were created to provide a control against the signaling effect during the cultivation step. Control plates for the cAMP-extracted suspensions consisted of only tenfold-diluted R2A plates. Control plates for plating AHL-treated suspensions had the same concentration of ethyl acetate as the concentration used during the extraction step.

## Cultivation and isolation of bacteria

After 1 day of agitation, each suspension was serially diluted with physiological solution (0.85% NaCl) by a factor of 10, $10^2$, $10^3$, $10^4$, and $10^5$. Dilutions were plated both on plates containing the same signaling molecule as in the extraction step and on control plates. After 3 days of cultivation at 28°C, the plates were washed with 3 mL of 0.85% NaCl solution with a hockey stick cell spreader. Two milliliters of material was recovered from the $10^6$ diluted plates, 200 µL from the $10^5$ diluted plates, and 20 µL from the $10^4$ diluted plates. These volumes were then pooled together into a single sample. In total, 36 samples were obtained, consisting of each combination (signaling molecule both in the extraction and plate, only in extraction, only in plate, or its solvent control, for the three signaling molecules, in triplicates).

## DNA isolation from soil suspensions and pooled samples

After the 24 hours agitation, 5 mL of the soil suspensions were dewatered by centrifugation (5,000 × $g$, 10 minutes). DNA was then isolated using a FastDNA Spin Kit for Soil (MP Bio, USA) according to the manufacturer's instructions. The pooled bacterial suspensions washed from plates were centrifuged in the same way. DNA was extracted from the resulting pellet using a PureLink Genomic DNA Minikit (Invitrogen, USA) according to the manufacturer's instructions (the gram-positive bacterial cell lysate protocol was followed). The hypervariable regions V4–V5 of the 16S rRNA genes were targeted using universal prokaryotic primers 515 forward (5′-GTGYCAGCMGCNGCGG-3′) and 926 reverse (5′- CCGYCAATTYMTTTRAGTTT-3′) (18). The PCR volume was 15 µL and contained: 7.5 µL KAPA HiFi HotStart ReadyMix (Kapa Biosystems, USA); 0.3 µM of each primer (Sigma-Aldrich, USA); and 60 ng of extracted DNA. The cycling program was set as follows: 5 minutes at 95°C, 20 cycles of 20 seconds at 98°C, 15 seconds at 56°C, 15 seconds at 72°C, and a final extension for 5 minutes at 72°C. A volume of 0.5 µL of the PCR product was used as the template for another PCR round, which was performed under the same conditions except that the final reaction volume was 25 µL, with a primer concentration of 1 µM (for each primer), and 10 cycles. The forward and reverse primers used for the second PCR were modified with sequencing adapters and internal barcodes

of variable length (5–8 bp) using a TaggiMatrix spreadsheet courtesy of Travis C. Glenn at the University of Georgia (https://baddna.uga.edu/). A mock community consisting of 15 bacterial strains was included as a positive control and amplified together with the samples (18). Further purification, amplicon-sample library preparation, and sequencing analysis in an Illumina MiSeq instrument were performed at the Core Facility for Nucleic Acid Analysis at the University of Alaska Fairbanks (AK, USA).

## Data analysis

The amplicon sequencing data were analyzed using DADA2 (19) in R (20). Primer sequences were trimmed off before the analysis. The sequences for each biological replicate were trimmed and filtered by their quality [truncLen = c(240,160), maxN = 0, maxEE = c (1, 1), truncQ = 2]. After dereplication, sequencing errors were removed (DADA2-based removal), denoised forward and reverse reads were merged, and chimeric sequences were removed. Taxonomy was assigned using the Silva ribosomal RNA gene database (21). Mitochondrial sequences and uncharacterized phyla were removed from both data sets. Low-abundance data were removed by filtering out amplicon sequence variants with abundances less than 70. Amplicon sequence variants (ASVs) coming from the pooled plates were separated from those coming from soil suspensions and were analyzed separately.

The analysis of the ASVs was carried out using the phyloseq package in R (22). ASVs were aligned using the package DECIPHER (23). The Bray-Curtis dissimilarity was used to evaluate the differences between the communities in soil and those growing in plates, as well as the community differences in the soil after the addition of the signaling molecules and the effect of the cultivation strategies. These dissimilarities were statistically analyzed through permutational analysis of variance (PERMANOVA) using the *Adonis2* function in the package Vegan (24). The diversity estimates were obtained using the estimate_richness function in Phyloseq. Statistical analyses were done in R using the packages Tidyverse (25) and rStatix (26). The false discovery rate method was used to correct *P* values in *post hoc* tests.

A differential expression analysis between different treatments for both the soil and the plate data sets separately was performed using the package metagenomeSeq (27). The data were agglomerated to the genus level before performing the analysis using the Phyloseq function "tax_glom". A zero-inflated Gaussian mixture model was fitted to the data using the function fitZig(). In the soil data set, the communities extracted with homoserine lactones were contrasted with the extraction with ethyl acetate only (control_solvent). Only genera with a false discovery rate (FDR) equal to or lower than 0.05 and a log2 fold change greater than 1.5 for the soil data set and 1 for the plate data set were considered as valid results of the analysis. The plates' ASVs were analyzed in the same way. Control plates (from control extractions and with no added signaling molecules) were contrasted against all the treatment possibilities (extraction with signaling molecules but plated on control plates; extraction with buffer alone but plated with signaling molecules; or extraction with signaling molecules and plated with signaling molecules). Bacteria that positively responded to the addition of signaling molecules in the soil data were searched for in the plate data. An increase in their abundance on plates was considered an indicator that the signaling molecules aided in their cultivation.

To predict the functions of the treated communities, the software PICRUSt2 (Phylogenetic Investigation of Communities by Reconstruction of Unobserved States) was run in Python with the ASV data generated from DADA2 (28). Briefly, the predictions in the form of functional orthologs (KEGG orthology numbers, KOs) resulting from the PICRUSt2 script were assigned descriptions using the add_descriptions.py script. These KOs were then grouped according to the metabolism they belonged to using the KEGG orthology database (29). Only KOs belonging to eight relevant metabolic functions were considered for further analysis: carbohydrate, energy, lipid, and nucleotide metabolism, cell motility, cellular community pathways of prokaryotes, membrane transport, and

signal transduction processes. Data visualizations were carried out using the package ggplot2 (30).

## RESULTS

### Effect of signaling molecules on soil communities

In total, 7,107,860 sequences were retrieved after the DADA2 pipeline processing, which accounted for 85.3% of the raw sequence data. The average number of sequences per sample was 67,055, with a minimum of 22,683 sequences. The overall data set had a final number of 1,061 individual ASVs after the removal of mitochondrial, uncharacterized phyla and low-abundance sequences, with the latter accounting for <1% of the valid sequences.

The communities of the soil extracts clustered separately in the ordination space (Fig. 1; MDS, Bray-Curtis dissimilarities, triplicate samples) and were significantly different from each other, as revealed by PERMANOVA (1,000 permutations, 6 levels, F = 6.52, $R2$ = 0.73, $d.f.1$ = 5, $d.f.2$ = 12, $P$ = 0.001). Each group's dispersion did not significantly vary across triplicates (betadisper, 6 levels, F = 0.51, $d.f.1$ = 5, $d.f.2$ = 12, $P$ = 0.79).

The effect of the signaling molecules on the community was also apparent when analyzing the diversity of each treated soil extract; the Shannon diversity index differed significantly between treatments (Fig. 2 and ANOVA, F = 63.97, $d.f.1$ = 5, $d.f.2$ = 12, $P$ < 0.05), with the highest diversity being observed in the original soil. *Post hoc* pairwise comparisons (FDR) showed that the original soil's Shannon diversity index was significantly higher than those of the other extractions ($P$ < 0.05) except for the control extraction with solvent. Whereas AHLs decreased the diversity compared to the extraction with solvent, the diversity after adding cAMP remained higher than the control extraction without solvent.

To determine which bacteria responded to each treatment, a differential abundance analysis was carried out. Supplementation with AHLs increased the abundance of several members of *Pseudomonadota* such as *Pseudomonas*, *Piscinibacter,* and the undescribed genus TX1A-55, as well as an undescribed genus of *Verrucomicrobiota* (Fig. 3). AHL1, in contrast to AHL2, influenced the relative abundance of *Afipia* and the undescribed genus Blyi10 (*Pseudomonadota*) while the addition of cAMP increased the relative abundance of the genus *Arsenicitalea*.

### Effect of signaling molecules on bacterial isolates

The diversity observed on plates, as represented by the abundance-based coverage estimators (ACE) index, significantly differed between the different extraction and plating strategies (Fig. 4 and ANOVA, F = 3.16, $d.f.1$ = 11, $d.f.2$ = 24, $P$ < 0.05); however, the structure of the communities did not (Fig. 5): PERMANOVA (1000 permutations, three levels, F = 7.18, $R2$ = 0.3, $d.f.1$ = 2, $d.f.2$ = 33, $P$ = 0.001) and dispersion differences between groups (betadisper, three levels, F = 2.34, $d.f.1$ = 2, $d.f.2$ = 33, $P$ = 0.11). The highest ACE index was observed for AHL2, when this signaling molecule was only added in the extraction step.

Of the treatments, the use of AHLs proved to be the most effective (Fig. 6, first to fifth panels) in terms of the fold change increase in certain taxa. Bacteria of the genera *Pseudomonas*, *Pseudoarthrobacter*, and *Paenarthrobacter* formed significantly more cells on R2A when AHLs were included in the extraction step. Additionally, AHL1 increased the abundance of *Nocardioides* cells, while cAMP increased the number of colonies of *Pseudarthrobacter* and *Pseudomonas* (Fig. 6). In the data set, 10 ASVs representing potentially novel species of *Nocardioides* and 1 of *Pseudomonas* were detected when using a threshold of 98.7%, which is used as a proxy for bacterial species delineation (31).

### Prediction of the community functions using PICRUSt2

PICRUSt2 revealed 1,517 different functional orthologs after the selection of eight categories relevant to bacterial metabolism. The relative abundance of these functional

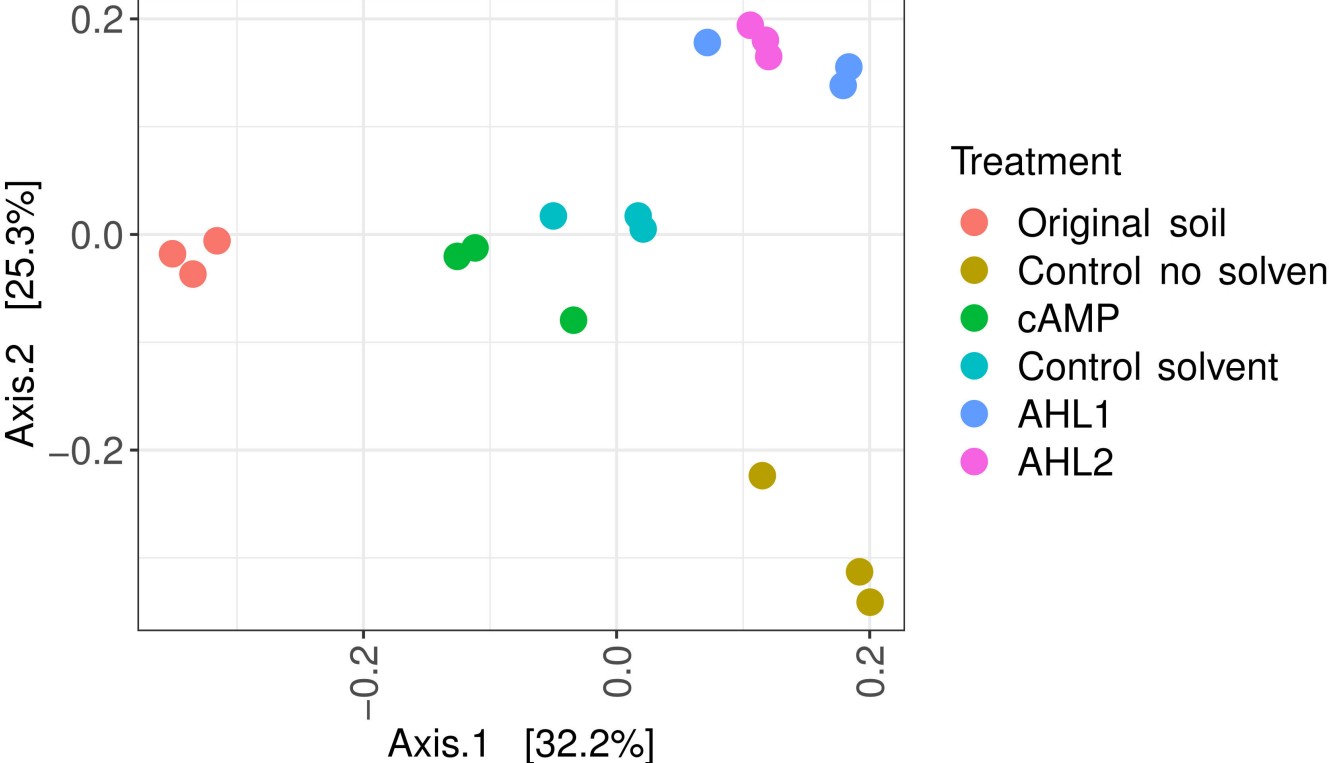

**FIG 1** Multidimensional scaling of Bray-Curtis dissimilarities between samples. Effect of signaling molecules, ethyl acetate-only extraction (Control_solvent), and buffer extraction (Control_no_solvent) on the composition of the communities in soil extracts. AHL1, N-(3-oxohexanoyl)-L-homoserine lactone; AHL2, N-octanoyl-L-homoserine lactone; cAMP, 3',5'-cyclic adenosine monophosphate.

orthologs is shown in Fig. 7. The addition of AHLs to the soil suspensions resulted in an increase in functions related to cell motility and cellular community, which are functions related to the use of AHLs in quorum sensing. Carbohydrate metabolism-related functions diminished when both AHLs were added.

## DISCUSSION

AHLs and cAMP belong to the array of signaling substances that bacteria are naturally exposed to in the environment. Soil presents specific challenges compared to the environments where these signaling molecules had been used before to increase cultivation (13, 15, 16, 32). Substances diffuse readily in aqueous environments such as lakes, oceans, or the human body but can have more difficulty moving in soil than in aqueous environments. Despite these possible diffusion difficulties, different AHLs triggered different responses in the soil community (Fig. 1 to 3).

The hydrophobicity of AHLs can also influence their adhesion to soil particles, which can impede their diffusion. In this study, the soil was turned first into a suspension so the diffusion of signaling molecules was facilitated. Differences in the community structure were identified after the addition of different signaling molecules (Fig. 1). In particular, AHL2 was more successful than cAMP or AHL1 at increasing the diversity of the community growing on plates (Fig. 4). Although AHLs are prone to degradation under different environmental conditions such as high pH or redox gradients (33), the neutral cultivation conditions during our experiments were not expected to result in a fast degradation of the signaling molecules, and the effect of the AHL addition was observed in the different community compositions (Fig. 1). The final concentration of signaling molecules in the soil suspensions was not measured.

The ACE diversity index calculated for the isolates (Fig. 4) suggests that of the two AHLs used in this study, AHL2 was more effective than AHL1 at increasing diversity on a

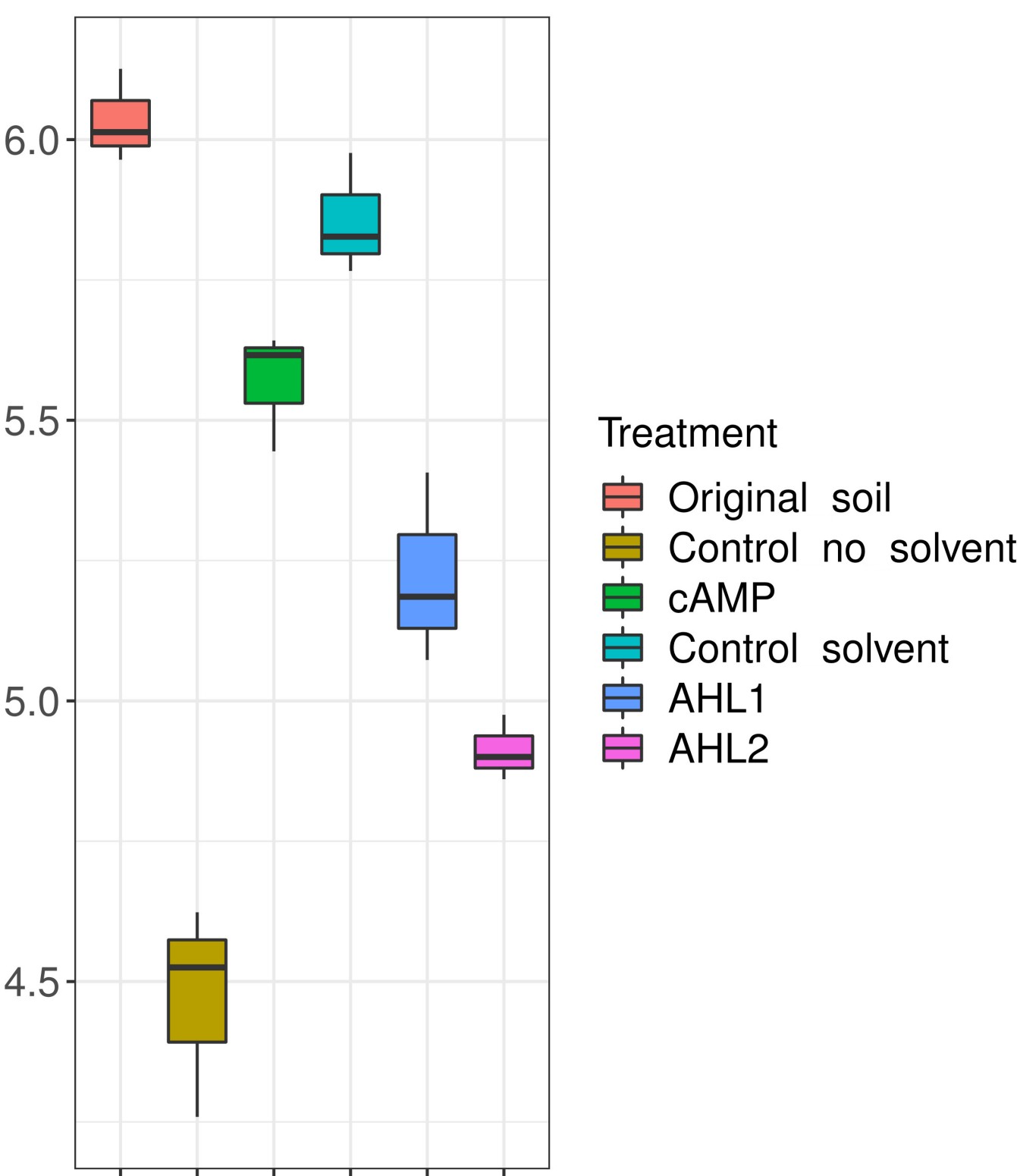

**FIG 2** Shannon index for original soil, control extractions, and extractions with signaling molecules. AHL1, N-(3-oxohexanoyl)-L-homoserine lactone; AHL2, N-octanoyl-L-homoserine lactone; cAMP, 3',5'-cyclic adenosine monophosphate.

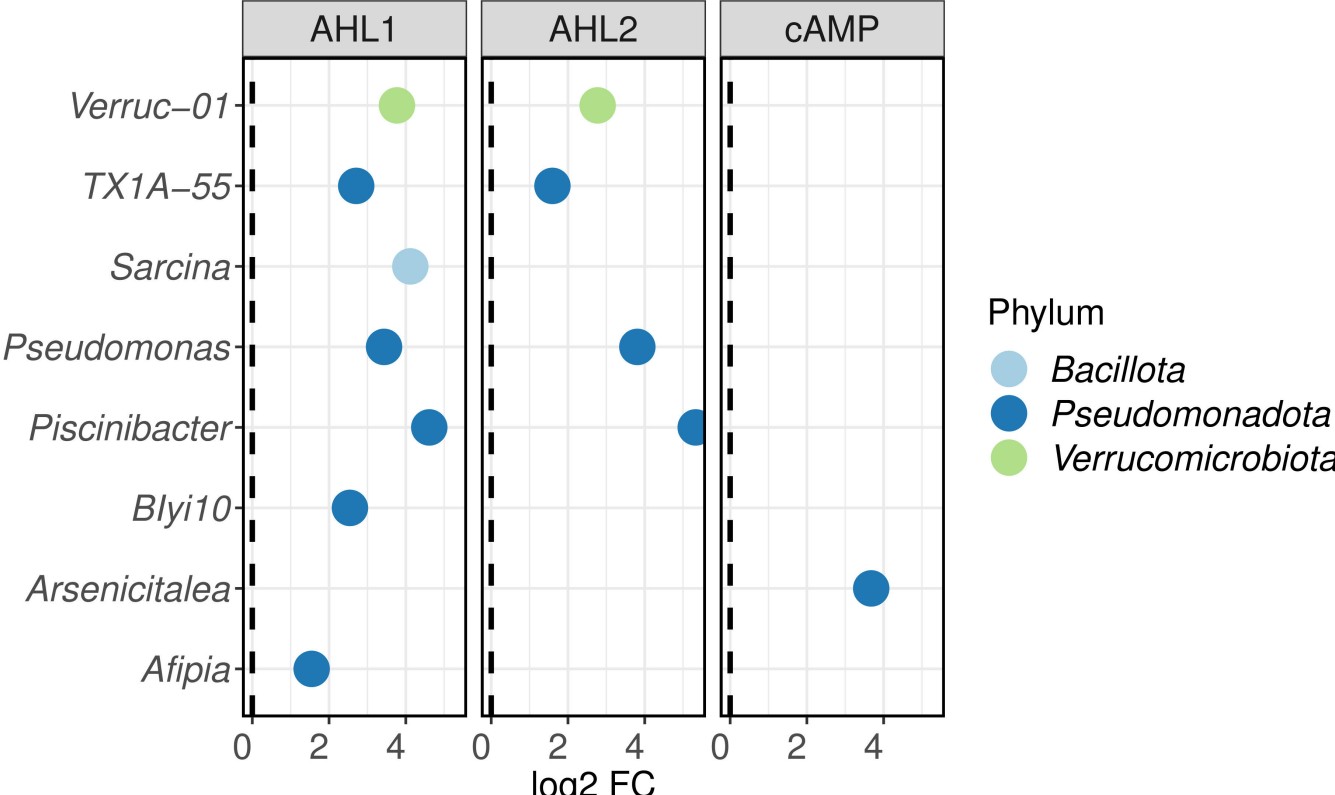

**FIG 3** Differential abundance analysis (metagenomeSeq) for different signaling molecules. Each treatment was compared to the extraction without a signaling molecule, either with solvent (for AHLs) or with no solvent [for 3',5'-cyclic adenosine monophosphate (cAMP)]. AHL1, N-(3-oxohexanoyl)-L-homoserine lactone; AHL2, N-octanoyl-L-homoserine lactone.

solid medium, and therefore the culturability of a larger number of organisms was increased on AHL2-treated plates compared to other treatments. The application of AHL2 in the extraction step proved particularly effective (Fig. 4 and 5). AHL2 is produced by many bacteria, including *Serratia* spp. (34, 35), *Citrobacter amalonaticus* (36), *Edwardsiella tarda* (37), *Rhizobium leguminosarum* (38), *Chromobacterium haemolyticum* (39), and *Burkholderia cepacia* (40). It is responsible for several bacterial responses such as virulence, the formation of biofilms, and swarming motility (40–42). It is also produced by the anammox *Planctomycetota* (43) and ammonium-oxidizing bacteria to promote nitrate oxidation (44). Finally, AHL2 is less polar than ALH1, so it could be a more important signal in the soil environment, where it can adhere to solid particles along with bacteria.

Specific taxa increased in abundance after the addition of signaling molecules (Fig. 3 and 6). A higher proportion of ASVs, particularly on plates (Fig. 6), suggests an increased culturability of those taxa on the medium used. A phylum for which the effect of the signaling molecules was very effective was *Pseudomonadota* (Fig. 3). Since AHLs are used as communication molecules, it is not surprising that other bacteria may have the ability to "eavesdrop," to listen to the "ongoing conversation". This eavesdropping capacity has been proposed to be possibly prevalent in *Pseudomonadota* (45). These signals could also be not just listened to but also intercepted and quenched by other bacteria. *Afipia* spp., for example, possess quorum quenching activity and have been observed to degrade AHLs in membrane bioreactors (46). AHLs in *Pseudomonas* are involved in important functions such as pigment production, motility, biofilm formation, and rhamnolipid production (47). *Piscinibacter*, which also belongs to *Pseudomonadota*, exhibits an increased expression of quorum sensing related genes during growth and colonization into aggregates (48). Isolates of all these bacteria, *Afipia*, *Pseudomonas*, and

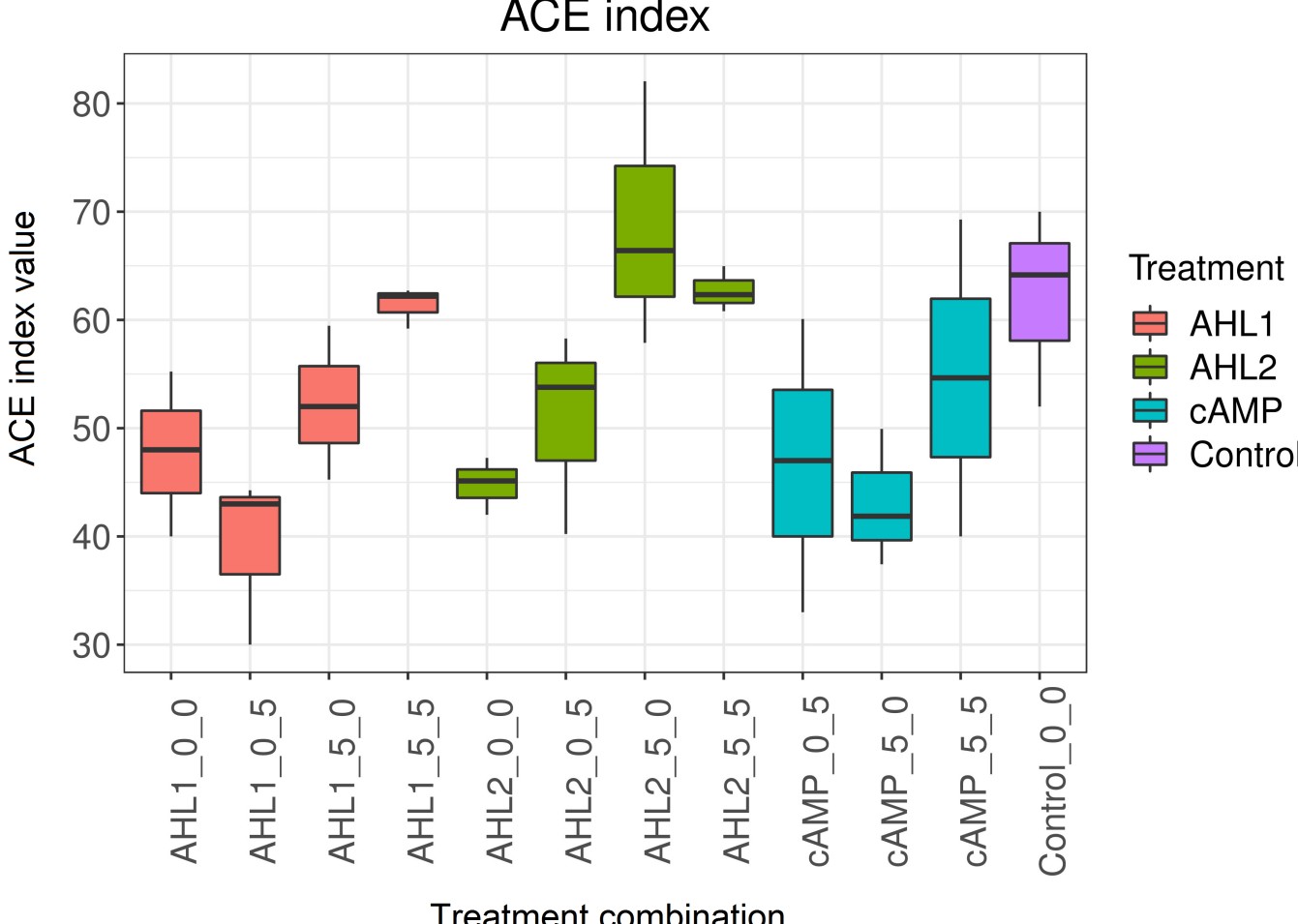

**FIG 4** Abundance-based coverage estimators (ACE) indexes of communities growing on plates. The format of the X-axis follows the pattern: signaling molecule _ concentration in extraction _ concentration in the solid medium. AHL1, N-(3-oxohexanoyl)-L-homoserine lactone; AHL2, N-octanoyl-L-homoserine lactone; cAMP, 3',5'-cyclic adenosine monophosphate.

*Piscinibacter*, increased in abundance after the addition of AHLs (Fig. 3), suggesting that these signaling molecules mainly benefit bacteria previously described to use AHL-related systems.

The abundance of a member of the phylum *Verrucomicrobiota* also increased after the addition of AHLs (Fig. 3). Similarly to members of *Pseudomonadota*, the abundance of *Verrucomicrobiota* has been shown to be correlated with AHL concentration in activated sludge (49). *Verrucomicrobiota*, unlike *Pseudomonadota*, is one of the phyla that despite the abundance of their members in natural environments is challenging to culture (50). Despite the fact that the relative abundance of sequences affiliated with *Verrucomicrobiota* increased in the treated soil extracts (Fig. 3), no isolates of this phylum have been cultured in this study. Such a result is in agreement with previous results (51), which shows that the supplementation of media with AHLs did not significantly increase the success of culturing *Verrucomicrobiota* as it did for *Acidobacteriota*.

Compared to AHLs, cAMP was less effective at increasing the abundance and thus the culturability of bacteria from soil (Fig. 3). This molecule was observed to aid the resuscitation and growth of starved bacteria and increase the culturability of both lake and lake sediment bacteria (13, 15, 32, 52). Because cAMP is an intracellular signal transductor, its role in extracellular communication could be more limited than that of homoserine lactones in soils.

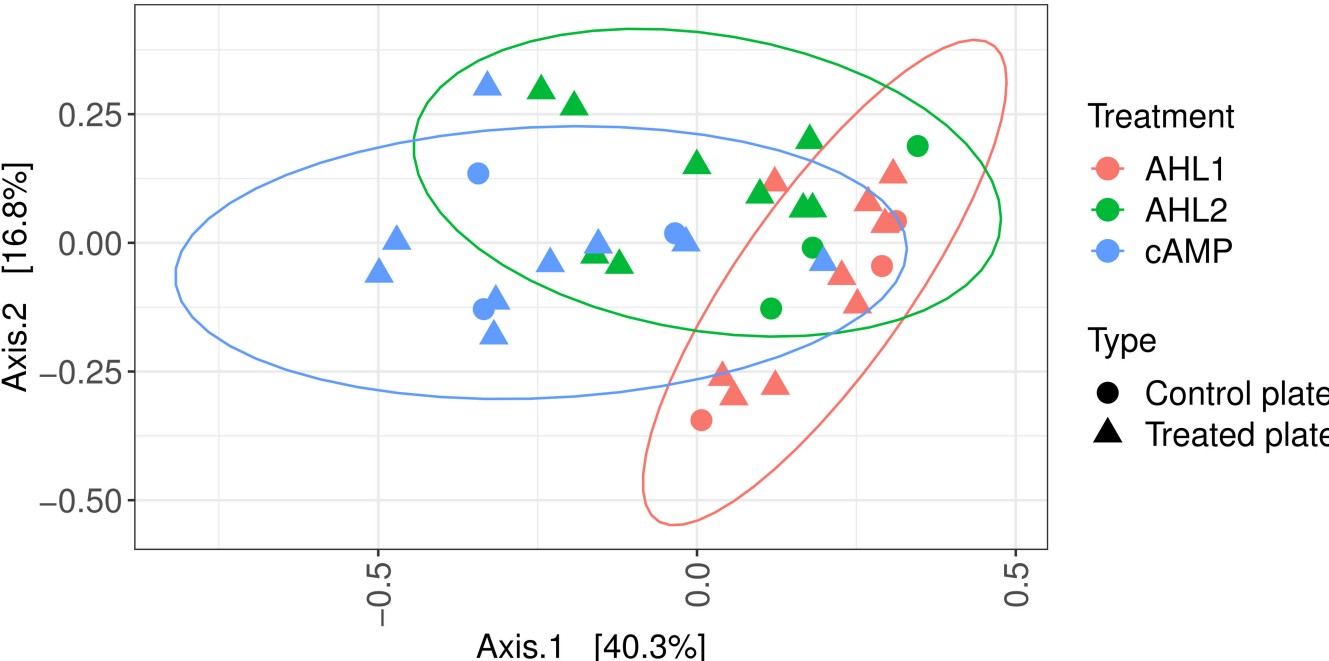

**FIG 5** Multidimensional scaling of Bray-Curtis dissimilarities between plate treatments. The figure shows the effect of each signaling molecule on the compositions of the communities growing on plates. The controls of each treatment are included within each treatment. Ellipses were drawn using the stat_ellipse function assuming a multivariate t-distribution. AHL1, N-(3-oxohexanoyl)-L-homoserine lactone; AHL2, N-octanoyl-L-homoserine lactone; cAMP, 3',5'-cyclic adenosine monophosphate.

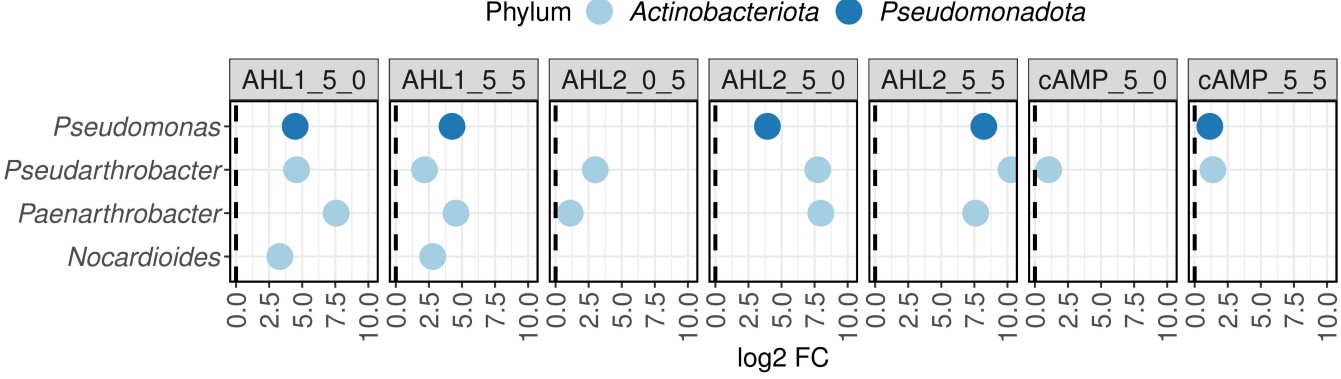

**FIG 6** Differential abundance analysis (metagenomeSeq) for plate ASVs. Each treatment and concentration level were compared to control plates with or without solvent. The format of the panel titles follows the pattern: signaling molecule_concentration during extraction_concentration in media. AHL1, N-(3-oxohexanoyl)-L-homoserine lactone; AHL2, N-octanoyl-L-homoserine lactone; cAMP, 3',5'-cyclic adenosine monophosphate.

The software PICRUSt2 was used to further look into the causes of the differentiation of the soil community after the extraction experiment (Fig. 1). Some orthologs of quorum sensing-related functions, such as cell motility and cellular community, were more abundant after the addition of AHLs than for any other treatment and control (Fig. 7). The community composition, shown in Fig. 1 and 3, also suggests that the action of both AHLs on the soil community was very similar. The same bacteria whose abundances increased after the addition of AHL2 also appeared with an increased abundance after the addition of AHL1 (Fig. 3), and both groups cluster together in the multidimensional scaling (Fig. 1).

The PICRUSt2 algorithm depends not only on the observed ASVs but also on their abundances (28). A drop in the abundance of carbohydrate metabolism-related orthologs is also observed in Fig. 7, first panel. A high concentration of signaling molecules

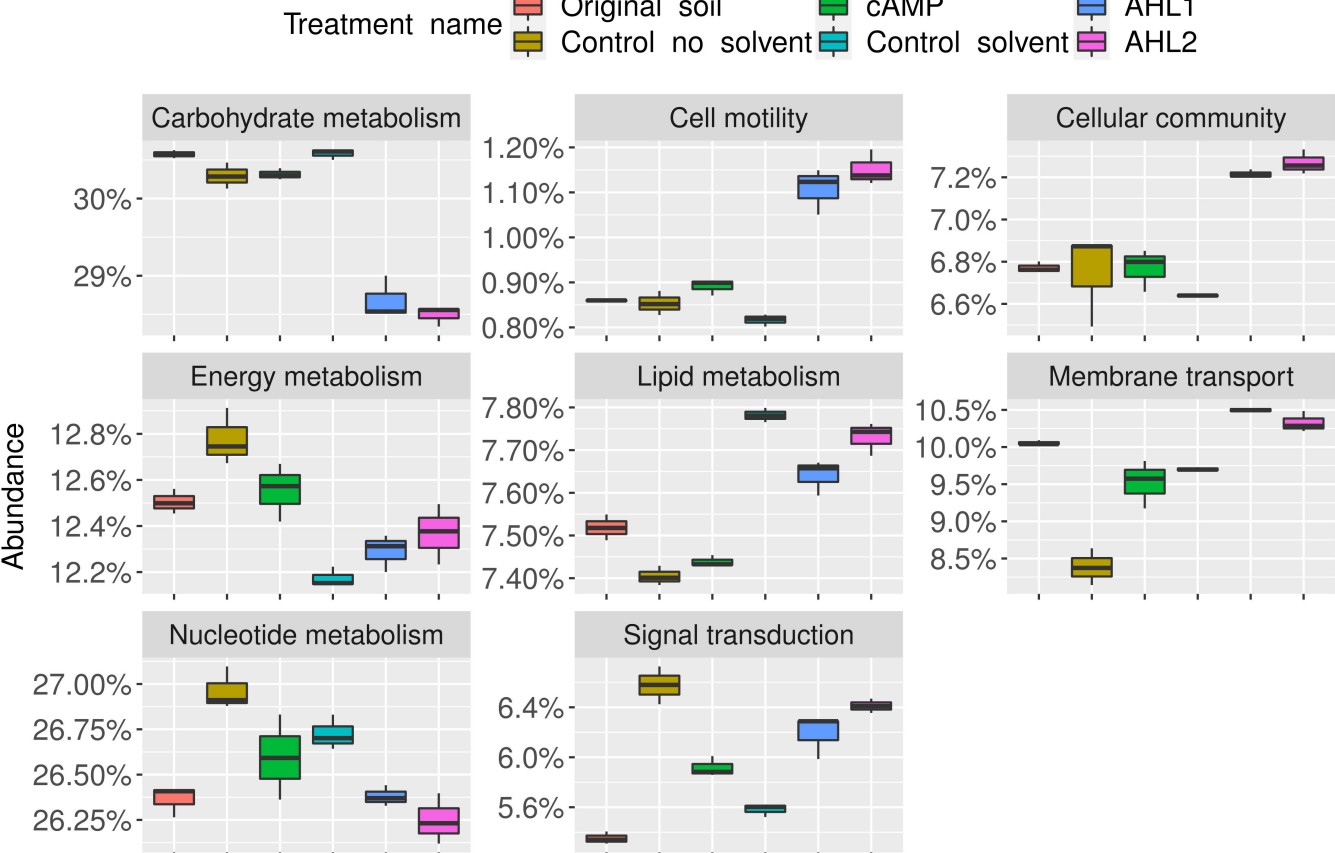

**FIG 7** Relative abundance of functional orthologs grouped according to eight relevant bacterial functions. AHL1, N-(3-oxohexanoyl)-L-homoserine lactone; AHL2, N-octanoyl-L-homoserine lactone; cAMP, 3',5'-cyclic adenosine monophosphate.

would suggest an overcrowded environment, so organisms can adapt and reduce their metabolism for a possible substrate shortage (53). PICRUSt2 predicts possible functions of a community and must not be taken as a proof of actual metabolic activity. For example, an increase in the abundance of *Pseudomonas* (Fig. 6) can mean that more genomes that encode functions such as those related to cell motility (Fig. 7) have been found by the software. Gene expression analyses or metatranscriptome sequencing would be needed to evaluate the real activities of the community after the addition of signaling molecules but were behind the scope of this manuscript.

In summary, this work shows that the addition of signaling molecules has the potential of increasing the culturability of soil bacteria under laboratory conditions. The addition of N-octanoyl-L-homoserine lactone was slightly more effective at increasing the diversity of bacteria on plates compared to the other signaling molecules used, N-(3-oxohexanoyl)-L-homoserine lactone and cAMP. Of the 69 sequences of genera with increased abundances after the signaling molecule treatment, 11 were members of potentially novel species. With that in mind, AHLs could be applied either in tandem with more high-throughput technologies, such as the ichip (54) or other high-throughput cultivation devices.

It is also important to note that the number of AHLs is large, so there is the possibility of increasing bacterial culturability using other AHLs. The two AHLs used in this study were selected on the basis of their reported ability to increase the growth rate and microbial activity of different bacterial lineages (55, 56). Other desired actions of AHLs can include enhancing metabolic activity, such as N-tetradecanoyl-DL-homoserine lactone or N-dodecanoyl-L-homoserine lactone in ammonium oxidizing bacteria (57), or inducing protection against oxidative stress, such as N-Decanoyl-DL-homoserine lactone

in *Burkholderia pseudomallei* (58). These increased capabilities could give several bacteria a higher chance to withstand the change of environment to laboratory conditions and support their growth there. Signaling molecules are thus a good tool to show bacteria that we understand their language and can convince them to grow more effectively in the laboratory environment.

## ACKNOWLEDGMENTS

The research reported here was supported by the Czech Science Foundation under grant no. 22-00132S and Ministry of Education, Youth and Sports of the Czech Republic under grant no. LTAUSA19028. Further support is acknowledged from the ELIXIR CZ research infrastructure project (Ministry of Education, Youth and Sports of the Czech Republic grant no. LM2023055), specifically for access to computing and storage facilities. M.A.L.M. further acknowledges the funding provided by the Mexican National Council of Humanities, Science and Technology (CONAHCYT).

## AUTHOR AFFILIATIONS

[1]Department of Biochemistry and Microbiology, University of Chemistry and Technology, Faculty of Food and Biochemical Technology, Prague, Czech Republic
[2]Department of Water Technology and Environmental Engineering, University of Chemistry and Technology Prague, Prague, Czechia

## AUTHOR ORCIDs

Marco A. Lopez Marin http://orcid.org/0000-0001-8546-7969
Ondrej Uhlik http://orcid.org/0000-0002-0506-202X

## FUNDING

| Funder | Grant(s) | Author(s) |
|---|---|---|
| Ministerstvo Školství, Mládeže a Tělovýchovy (MŠMT) | LTAUSA19028 | Ondrej Uhlik |
| Ministerstvo Školství, Mládeže a Tělovýchovy (MŠMT) | LM2023055 | Ondrej Uhlik |
| Grantová Agentura České Republiky (GAČR) | 22-00132S | Ondrej Uhlik |

## AUTHOR CONTRIBUTIONS

Marco A. Lopez Marin, Data curation, Formal analysis, Investigation, Methodology, Software, Validation, Visualization, Writing – original draft | Michal Strejcek, Funding acquisition, Resources, Supervision, Writing – review and editing | Ondrej Uhlik, Conceptualization, Funding acquisition, Project administration, Resources, Supervision, Writing – review and editing

## DATA AVAILABILITY

The data obtained in this work were deposited in the NCBI Short Read Archive under BioProject accession number PRJNA943488.

## ADDITIONAL FILES

The following material is available online.

Open Peer Review

**PEER REVIEW HISTORY (review-history.pdf).** An accounting of the reviewer comments and feedback.

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
