## [Reviewer comments · Microbiology Spectrum]

Microbiology Spectrum

Joining the bacterial conversation: increasing the cultivation efficiency of soil bacteria with acyl-homoserine lactones and cAMP

Marco Lopez Marin, Michal Strejcek, and Ondrej Uhlík

Corresponding Author(s): Marco Lopez Marin, Vysoka skola chemicko-technologicka v Praze

Review Timeline:

Submission Date:	May 3, 2023
Editorial Decision:	June 25, 2023
Revision Received:	August 16, 2023
Accepted:	August 17, 2023

Editor: Jeffrey Gralnick

Reviewer(s): Disclosure of reviewer identity is with reference to reviewer comments included in decision letter(s). The following individuals involved in review of your submission have agreed to reveal their identity: Tatyana L. Povolotsky (Reviewer #2)

Transaction Report:

DOI: <https://doi.org/10.1128/spectrum.01860-23>

June 25, 2023

Dr. Marco Antonio Lopez Marin
Vysoka skola chemicko-technologicka v Praze
Department of Biochemistry and Microbiology
Technicka 3
Prague 16628
Czech Republic

Re: Spectrum01860-23 (Joining the bacterial conversation: increasing the cultivation efficiency of soil bacteria with acyl-homoserine lactones and cAMP)

Dear Dr. Marco Antonio Lopez Marin:

Thank you for submitting your manuscript to Microbiology Spectrum. Your manuscript was reviewed by two experts in the field, but only Reviewer 1 provided substantial comments, which you will find below. One major point for your consideration is regarding the quantification of AHL in soils. I would also suggest that discussion of the metagenomic prediction (Fig 7) results be carefully revised. The software is making an educated guess to metabolic functionality and may or may not reflect reality (please also address this point in your response to reviewers). You may use it if you wish, but the limitations of this approach should be discussed.

Link Not Available

Sincerely,

Jeffrey Gralnick

Journals Department
Reviewer comments:

Reviewer #1 (Comments for the Author):

The manuscript by authors explained the signaling molecules AHLs which play an important role in the interaction of bacteria

especially Gram-negative. Though this study is interesting, detailed information is missing in the manuscript that must be performed carefully since some of the information seems superficial. The author should clearly confirm the AHLs concentration from soil too as compared to soil microbial communities.

General comments

1. The proper concentration of AHLs applied in the soil is missing in the manuscript, that is a major drawback of this manuscript. 5 μm is much less that can provoke the signaling molecule inside the soil.
2. The AHL signaling molecules are activated in the presence of special reported strains like *Agrobacterium tumefaciens* (Waheed et al., 2022), that is another missing point in the study.
3. The addition of these chemicals in the soil needs proper environment for reaction that can manipulate the soil microbial communities.
4. Since the authors concluded that the addition of AHLs molecule can increase the culture ability under laboratory conditions, none of the experiments are present to proof this.
5. Did the author finally confirm the recovery of total AHL after addition to the soil through LC/MS.MS or other techniques? This is very important information as far as AHLs are concerned.

Reviewer #2 (Comments for the Author):

The paper was well written, interesting to the scientific community and in the scope of this journal. Has good future research potential. Was logical and easy to follow.

Staff Comments:

Preparing Revision Guidelines

Please return the manuscript within 60 days; if you cannot complete the modification within this time period, please contact me. If you do not wish to modify the manuscript and prefer to submit it to another journal, please notify me of your decision immediately so that the manuscript may be formally withdrawn from consideration by Microbiology Spectrum.

June 25, 2023

Dr. Marco Antonio Lopez Marin
Vysoka skola chemicko-technologicka v Praze
Department of Biochemistry and Microbiology
Technicka 3
Prague 16628
Czech Republic

Re: Spectrum01860-23 (Joining the bacterial conversation: increasing the cultivation efficiency of soil bacteria with acyl-homoserine lactones and cAMP)

Dear Dr. Marco Antonio Lopez Marin:

Thank you for submitting your manuscript to Microbiology Spectrum. Your manuscript was reviewed by two experts in the field, but only Reviewer 1 provided substantial comments, which you will find below. One major point for your consideration is regarding the quantification of AHL in soils. I would also suggest that discussion of the metagenomic prediction (Fig 7) results be carefully revised. The software is making an educated guess to metabolic functionality and may or may not reflect reality (please also address this point in your response to reviewers). You may use it if you wish, but the limitations of this approach should be discussed.

<https://spectrum.msubmit.net/cgi-bin/main.plex?el=A6QF5CJXd2A3FcGr3F2A9ftdB5S8oPr2MSCiySshPA9SOwZ>

ASM policy requires that data be available to the public upon online posting of the article, **so please verify all links to sequence records**, if present, and make sure that each number retrieves the full record of the data. If a new accession number is not linked

or a link is broken, provide production staff with the correct URL for the record. If the accession numbers for new data are not publicly accessible before the expected online posting of the article, publication of your article may be delayed; please contact the ASM production staff immediately with the expected release date.

Sincerely,

Jeffrey Gralnick

Journals Department
The limitations of the PICRUSt2 software were discussed as suggested. The new discussion comprises lines 335 to 341 of the corrected manuscript.

Reviewer comments:

Reviewer #1 (Comments for the Author):

The manuscript by authors explained the signaling molecules AHLs which play an important role in the interaction of bacteria especially Gram-negative. Though this study is interesting, detailed information is missing in the manuscript that must be performed carefully since some of the information seems superficial. The author should clearly confirm the AHLs concentration from soil too as compared to soil microbial communities.
General comments

1. The proper concentration of AHLs applied in the soil is missing in the manuscript, that is a major drawback of this manuscript. 5 μ m is much less that can provoke the signaling molecule inside the soil.

AHLs were included to a final concentration of 5 μM , as specified in lines 104-105. Following your comment, the used concentration was also added to the abstract in line 21. In previous studies using AHLs, for example those of Bruns et al, similarly low concentrations of signalling molecules (10 μM) were enough to elicit a greater number of viable cells. To emphasize that the concentration used actually had an effect on the composition of the soil microbial community, a new sentence was included in lines 266-267.

2. The AHL signaling molecules are activated in the presence of special reported strains like *Agrobacterium tumefaciens* (Waheed et al., 2022), that is another missing point in the study.

Agrobacterium tumefaciens is commonly used as a bioreporter to detect the presence of AHLs, and this was the function it played in Waheed's study. As also answered in comment #5, in our study, the presence or amount of signalling molecules was not determined after the 24 hours of extraction with PBS. We understand that this methodology is missing, either using *A. tumefaciens* or LC/MS.MS as mentioned in question 5, and this has been made explicit in lines 277-278.

3. The addition of these chemicals in the soil needs proper environment for reaction that can manipulate the soil microbial communities.

It is true that AHLs can be rendered inactive in nature under environmental conditions such as basic pH, but even in very basic environments, AHLs have been found (Decho et al, 2011). The higher the molecular weight of the AHLs used, the slower their hydrolysis (Decho et al 2011). For instance, for an AHL with 14 carbons, the half hydrolyzation rate was 1000 min, so a total degradation of the AHLs in the 24 hours that the experiment lasted is not expected. Moreover, AHLs would be more susceptible in natural systems with changing physicochemical parameters. In our case, PBS was added as buffer, to ensure constant conditions. These conditions were not extreme in terms of pH (pH used = 7.4) or temperature, so a fast degradation of the AHLs added was not expected.

This explanation was included in the text in lines 273-277. The pH of the buffer was added in line 99.

4. Since the authors concluded that the addition of AHLs molecule can increase the culture ability under laboratory conditions, none of the experiments are present to proof this.

Two analyses were done to assess the culturability increase after the addition of signalling molecules: the increase in alpha diversity observed on plates after the addition of the molecules (figure 4) and the differential increase of particular taxa after the addition of signalling molecules (figures 3 and 6). An increase in the alpha diversity of the

community growing on plates means that more organisms are growing on the plate compared to the control. This was highlighted in lines 281-282 and 291-293.

5. Did the author finally confirm the recovery of total AHL after addition to the soil through LC/MS.MS or other techniques? This is very important information as far as AHLs are concerned.

The recovery of total AHL after addition to soil was not confirmed, and therefore not compared to the AHL concentration in the native soil. This information cannot be obtained now, in particular because around two years have already elapsed since the experiments were done. The soil continues in refrigeration, but the community will not be the same as it was when the experiment was done. The lack of a recovery methodology was made explicit in lines 277-278. Our hypothesis is restricted to the use of signalling molecules to increase the number of bacteria growing on a solid medium (mainly shown in figure 4). We agree with this reviewer that the total recovery would be indeed good support for the study, so would be, for instance, gene expression analyses or metabolic rate measurements, etc. However, our data are convincing enough that our hypothesis can be accepted with the experimental design we used in this study. Moreover, previous experiments dealing with the use of signalling molecules for increasing the cultivation efficiency of bacteria have not dealt with their residual amounts (Bruns et al, 2002, 2003).

Reviewer #2 (Comments for the Author):

The paper was well written, interesting to the scientific community and in the scope of this journal. Has good future research potential. Was logical and easy to follow.

We thank the reviewer for the comment.

August 17, 2023

Dr. Marco Antonio Lopez Marin
Vysoka skola chemicko-technologicka v Praze
Department of Biochemistry and Microbiology
Technicka 3
Prague 16628
Czech Republic

Re: Spectrum01860-23R1 (Joining the bacterial conversation: increasing the cultivation efficiency of soil bacteria with acyl-homoserine lactones and cAMP)

Dear Dr. Marco Antonio Lopez Marin:

Your manuscript has been accepted, and I am forwarding it to the ASM Journals Department for publication. You will be notified when your proofs are ready to be viewed.

Sincerely,

Jeffrey Gralnick
Editor, Microbiology Spectrum
